# Ribosomal Protein uS5 and Friends: Protein–Protein Interactions Involved in Ribosome Assembly and Beyond

**DOI:** 10.3390/biom13050853

**Published:** 2023-05-18

**Authors:** Anne-Marie Landry-Voyer, Zabih Mir Hassani, Mariano Avino, François Bachand

**Affiliations:** Dept of Biochemistry & Functional Genomics, Université de Sherbrooke, Sherbrooke, QC J1E 4K8, Canada; anne-marie.landry-voyer@usherbrooke.ca (A.-M.L.-V.); zabihullah.mir.hassani@usherbrooke.ca (Z.M.H.); mariano.avino@usherbrooke.ca (M.A.)

**Keywords:** uS5, dedicated chaperone, PDCD2, PDCD2L, PRMT3, ZNF277, ribosome biogenesis

## Abstract

Ribosomal proteins are fundamental components of the ribosomes in all living cells. The ribosomal protein uS5 (Rps2) is a stable component of the small ribosomal subunit within all three domains of life. In addition to its interactions with proximal ribosomal proteins and rRNA inside the ribosome, uS5 has a surprisingly complex network of evolutionarily conserved non-ribosome-associated proteins. In this review, we focus on a set of four conserved uS5-associated proteins: the protein arginine methyltransferase 3 (PRMT3), the programmed cell death 2 (PDCD2) and its PDCD2-like (PDCD2L) paralog, and the zinc finger protein, ZNF277. We discuss recent work that presents PDCD2 and homologs as a dedicated uS5 chaperone and PDCD2L as a potential adaptor protein for the nuclear export of pre-40S subunits. Although the functional significance of the PRMT3–uS5 and ZNF277–uS5 interactions remain elusive, we reflect on the potential roles of uS5 arginine methylation by PRMT3 and on data indicating that ZNF277 and PRMT3 compete for uS5 binding. Together, these discussions highlight the complex and conserved regulatory network responsible for monitoring the availability and the folding of uS5 for the formation of 40S ribosomal subunits and/or the role of uS5 in potential extra-ribosomal functions.

## 1. Introduction

Despite the expanding number of roles played by noncoding RNAs, proteins remain key actors involved in nearly every operation required for cellular life, from proliferation to differentiation, internal organization, intercellular communication, and cell death. In order to set in motion the synthesis of new proteins, the information encoded by genes as messenger RNAs (mRNAs) is decoded into polymers of amino acids by a highly complex cellular machine, the ribosome, in a process known as translation. The ribosome is one of the most important ribonucleoprotein complexes in the cell, as demonstrated by its essential role in protein synthesis, its highly conserved nature, and its dominating abundance in most cell types. In fact, the fundamental structure and function of the ribosome were highly conserved throughout the evolution from bacteria to humans [1,2]. Since the topic of this review will focus on eukaryotes, the following paragraphs will refer to the eukaryotic ribosome unless otherwise indicated.

The 80S ribosome is a large RNA–protein complex with a molecular mass of 4.3 megadalton in humans [3] and is composed of two independent subunits: the 40S (or small) and 60S (or large) ribosomal subunits. The 40S ribosomal subunit consists of 33 different ribosomal proteins (RPs) and a single ribosomal RNA (rRNA), the 1869-nt-long 18S rRNA, whereas the 60S ribosomal subunit contains 47 RPs and three different RNA molecules: the 5S (121-nt), 5.8S (157-nt), and 28S (5070-nt) rRNAs [3]. While the 40S subunit contains the decoding center that monitors the complementarity of mRNA and tRNA during translation, the peptidyl-transferase center and the exit tunnel, in which the nascent polypeptide emerges out the ribosome, are at the heart of the 60S ribosomal subunit [4].

The synthesis of new ribosomes is one of the most energy demanding and complex processes occurring in eukaryotic cells. In addition to the four rRNAs and 80 RPs, ribosome biogenesis involves the coordinated action of the three cellular RNA polymerases, several hundred ribosome biogenesis factors (RBFs), as well as about 200 small nucleolar RNAs (snoRNAs) [5]. Ribosome synthesis begins in the nucleolus, where nascent rRNA is transcribed by RNA polymerase I (RNAPI) and assembled co-transcriptionally into a 90S pre-ribosomal particle via the spatio-temporal recruitment of several RPs, RBFs, and snoRNPs. Following endonucleolytic cleavage of the primary transcript between 18S and 5.8S rRNAs sequences, pre-40S and pre-60S particles will subsequently follow distinct maturation pathways. Whereas this endonucleolytic cleavage step mainly occurs co-transcriptionally in budding yeast [6], the extent to which this internal cleavage step is co-transcriptional in mammalian cells remains unclear. RNAPIII transcribes the fourth rRNA, the 5S rRNA, which joins the pre-60S particle in the nucleolus as part of the 5S RNP complex [7]. After transiting through the nucleoplasm, pre-40S and pre-60S particles are independently exported to the cytoplasm where they will be further processed to ultimately become competent for translation [5,7,8].

While the main role of the 80 RPs is to assist in the folding of the four rRNA molecules into a three-dimensional structure required for the precise interaction of mRNA codons with tRNA anticodons, the coordinated incorporation of RPs into their corresponding pre-ribosomal particle is critical for the stepwise assembly of mature ribosomal subunits. Specifically, functional studies in yeast and human cells show that deficiency of most RPs affects specific steps of pre-rRNA processing associated with pre-90S, pre-40S, and/or pre-60S maturation, which usually coincides with the timing of RP incorporation into pre-ribosomes [9,10,11,12]. Accordingly, most genes that code for RPs are essential for cellular proliferation and viability as well as for embryonic development in multi-cellular organisms [4].

The eukaryotic ribosomal protein uS5 (also referred as RPS2), which is homologous to the prokaryotic 30S ribosomal protein S5 (RPS5/rpsE), is one of the largest RPs of the 40S ribosomal subunit. Interestingly, the past 20 years has seen the identification of several evolutionarily conserved uS5-associated proteins. However, the biological significance of the interaction of uS5 with many of these proteins remains to be defined. In this review, we begin by revising the knowledge on the functional role of uS5 in the late stages of pre-40S maturation as well as evidence supporting that uS5 contributes to translation fidelity. We next outline the considerable list of conserved uS5-associated proteins and discuss their functions in ribosome biogenesis and beyond.

## 2. Structural Features of Eukaryotic uS5 and Role in Translation

Human uS5 is a 293-amino-acids-long protein with a molecular mass of approximately 31 kDa that shows cytoplasmic expression in most tissues [13]. Analyses of actively translating ribosomes by cryo-electron microscopy (cryo-EM) [14] reveal that uS5 is located on the solvent-exposed side of the 40S ribosomal subunit (Figure 1A). Specifically, in the context of the mature 40S ribosomal subunit, uS5 is physically connected with ribosomal proteins eS21, uS2, uS8, uS4, and uS3. Residues of uS5 also interact with the 18S rRNA via a double-stranded RNA-binding-like domain (Figure 1B, magenta; PROSITE entry PS50881) and the conserved S5 C-terminal domain (Figure 1B, orange; PROSITE entry PS00585). Like many other ribosomal proteins, uS5 adopts a globular structure that is associated with disordered N- and C-terminal extensions [4,14] (Figure 1B). More precisely, the first 56 and last 14 amino acids of uS5 were not modeled from the cryo-EM analysis of active ribosomes [14] and show very low structural confidence scores as predicted by AlphaFold [15], consistent with disordered regions (Figure 1B). Notably, both N- and C-terminal extensions are unique to eukaryotic uS5 and absent in the *E. coli* homolog (Figure 1C). As shown in Figure 1D, the eukaryotic-specific N-terminal extension of uS5 is rich in arginine and glycine residues. Arginine-glycine (RG)-rich motifs have been associated with mediating interactions with RNA and protein as well as contributing to nuclear localization [16]. Interestingly, several arginine residues in the N-terminal RG-rich extension of uS5 are targeted by asymmetric dimethylation (see section on PRMT3), a uS5 post-translational modification that appears to be evolutionarily conserved [17,18,19]. Finally, human uS5 would include an unconventional nuclear localization signal (NLS), between lysine-159 and threonine-232, which would allow uS5 to be transported to the nucleus by various import receptors [20].

uS5 has been shown to be important for translation fidelity in *E. coli*, especially for a conserved glycine at position 28 [21,22,23,24]. *E. coli* uS5, together with uS3 and uS4, form part of the tunnel through which mRNA enters the small subunit of the ribosome to reach the interface between the two subunits [25]. While uS3 and uS4 act as RNA helicases, uS5 orients the incoming mRNA for proper codon reading in the ribosome A site [26]. In eukaryotes, based on cryo-EM structures of the yeast pre-initiation complex following AUG recognition [27], recent findings using *Saccharomyces cerevisiae* also support a role for eukaryotic uS5 in translation fidelity, especially at the level of translational initiation [28]. Accordingly, substitutions of uS5 residues identified as proximal to mRNA nucleotides 8 to 13 downstream of the AUG start codon [27] were shown to enhance translation initiation at suboptimal start codons [28], suggesting that uS5–mRNA contacts may contribute to the stability and thermodynamics of the eukaryotic preinitiation complex. Recent studies in mammals also support the role of uS5 in translation fidelity, as a substitution of alanine-226 for a tyrosine in uS5 leads to increased mistranslation in human cells [29] and muscle atrophy in mice [30].

## 3. uS5 Is an Essential Protein Required for 40s Ribosomal Subunit Production

Whereas most yeast RPs are encoded by duplicated paralogous genes, uS5 is one of the few RPs encoded by a single gene in both budding and fission yeast. *uS5* is an essential gene in budding yeast as its deletion in *S. cerevisiae* yields inviable spores [31]. Accordingly, uS5 expression is required for ribosome biogenesis. A conditional mutant strain of uS5 in *S. cerevisiae* results in the accumulation of 20S rRNA precursors; yet, it also shows a reduction in newly made 20S pre-rRNA molecules in the cytoplasm, suggesting a role for uS5 in the export of pre-40S particles [10]. As for budding yeast, uS5 is also essential for cell viability in fission yeast, and knockdown of uS5 results in the complete inhibition of 40S ribosomal subunit production [32]. Notably, *Schizosaccharomyces pombe* cells depleted of uS5 showed only a small fraction of pre-rRNA matured into 20S precursors, suggesting that a large fraction of pre-40S is actively turned-over in the absence of uS5 [32]. In *Drosophila*, *uS5* was identified as the allele associated with the “string of pearl (sop)” recessive female sterile mutants [33]. The *sop* allele is associated with reduced *uS5* mRNA levels, oogenesis and early development defects, larval lethality, and a Minute-like phenotype [33]. The Minute syndrome in *Drosophila*—which includes delayed development, low fertility and viability, and decreased body size—is thought to arise as a consequence of suboptimal protein synthesis that results from reduced levels of cellular ribosomes [34]. In mammals, most of our knowledge about uS5 comes from studies performed on immortalized cell lines. Consistent with findings in yeast and *Drosophila*, *uS5* is an essential gene in most tested human cancer cell lines [35], thereby making uS5 a potentially interesting target for cancer vulnerabilities [36]. Biochemical and structural data obtained from human cells indicate that uS5 is incorporated at late stages of pre-40S particle assembly prior to nuclear export [11,37]. Accordingly, knockdown of uS5 in human cell lines results in the accumulation of 21S pre-rRNA, suggesting the uncoupling of cleavage at sites A0–1 in the 5′ external transcribed spacer sequence [11,38], as well as increase detection of 18S-E precursors in the nucleus, consistent with delayed nuclear export of pre-40S particles [11].

Collectively, the current data support a conserved role for uS5 in the late stages of 40S ribosomal subunits assembly. Consistent with this conclusion, recent cryo-EM analyses of pre-40S intermediates isolated prior to nuclear export suggest that uS5 is incorporated into nucleoplasmic pre-40S particles [37]. Interestingly, although data generally support that the ribosome assembly process is largely conserved between yeast and human cells [7], recent results suggest that uS5 may incorporate pre-40S particles at different time points between yeast and humans [37]. Specifically, *S. cerevisiae* uS5 was detected in pre-40S particles before the incorporation of uS2 and eS21, whereas, in human cells, the timing of uS5 incorporation coincided with the insertion of uS2 and eS21, suggesting that uS2–uS5–eS21 are incorporated as a cluster in humans [37].

### A Multifaceted Network of uS5-Associated Proteins

The identification of evolutionarily conserved uS5-associated proteins has been the focus of several studies in the past two decades. Such studies have provided new insights into the processes and mechanisms that promote uS5 expression and incorporation into ribosomes, as well as possible yet-to-be-defined extra-ribosomal functions. The next sections will focus on the best-characterized and -conserved uS5-associated proteins: PDCD2, PDCD2L, PRMT3, and ZNF277.

## 4. PDCD2 and PDCD2L: uS5-Associated Paralogs That Take Part in Human Ribosome Assembly

During the process of establishing that the uS5–PRMT3 complex, which was originally identified in fission yeast [17], is conserved in humans, a set of novel and highly specific PRMT3 interactors were identified in addition to uS5, including strong enrichments of the PDCD2 and PDCD2-like (PDCD2L) proteins [39]. Biochemical assays further demonstrated that uS5 bridges the association between PRMT3 and PDCD2/PDCD2L, as depletion of uS5 totally prevented the copurification of PRMT3 and PDCD2/PDCD2L [39]. *PDCD2* and *PDCD2L* are paralogous genes conserved through evolution, with homologs from bacteria to animals but not in archaebacteria. Based on sequence analysis, *PDCD2* is thought to have arisen from the duplication of the *PDCD2L* gene prior to the divergence of animals, fungi, and plants from a common ancestor [40]. Homologs of human PDCD2 and PDCD2L paralogs are also found in mice (Pdcd2 and Pdcd2l), in *Drosophila* (Zfrp8 and Trus), and in fission yeast (Trs401 and Trs402); however, a single homolog is found in budding yeast (Tsr4). As shown in Figure 2A, PDCD2 and PDCD2L (34% identical; 52% similar) belong to a family of proteins containing N- and C-terminal TYPP (Tsr4, YwqG, PDCD2L, PDCD2) domains [40], each consisting of GGxP and Cx_1-2_C-like motifs as well as a highly conserved glutamine (Q) residue (see Figure 2A). In PDCD2, the N- and C-terminal TYPP motifs are interrupted by the insertion of a MYND-type zinc finger, which was shown to be involved in transcriptional repression via protein–protein interactions [41,42]. On the other hand, PDCD2L lacks the MYND zinc finger but contains a leucine-rich nuclear export sequence (NES) that enables PDCD2L to exit the nucleus in a CRM1-dependent manner [39] (Figure 2A). While the MYND domain is conserved in *Drosophila* Zfrp8 (Figure 2B), it is not found in the *S. cerevisiae* homolog of PDCD2 (Tsr4). The C-terminal TYPP domain also appears to be degenerated in yeast Tsr4 (Figure 2B, note lack of Cx_1-2_C motif), resulting in a predicted structure that is markedly different from other PDCD2 homologs (Figure 2C). The functional role of the TYPP domain has not been well studied, though it is thought to facilitate chaperoning activity and protein–protein interactions [40]. Indeed, substitutions that modify key residues conserved in the TYPP domain of PDCD2 completely abolish the association between PDCD2 and uS5 in human cells [38].

## 5. PDCD2 Is a Conserved Dedicated Chaperone for uS5

The *PDCD2* gene was originally identified in a screen for mRNAs upregulated upon apoptosis in rat cells [43]. However, subsequent experiments failed to support a correlation between *PDCD2* mRNA expression and apoptosis [44,45]. Since then, PDCD2 has been associated with the pathogenesis of several disorders, including cancer [44,46,47,48,49,50,51,52], Parkinson’s disease [53], and fragile X syndrome [54]. Along with its potential role in diseases, PDCD2 is also implicated in development. In mice, PDCD2 is essential for stem cell viability and proliferation, and its absence leads to early embryonic lethality [55]. Although the aforementioned studies establish a clear role for PDCD2 in the development of human disorders as well as during embryonic development, the molecular function of PDCD2 had remained largely elusive until recently. Indeed, a set of elegant studies in budding yeast and human cell lines both support the conclusion that Tsr4/PDCD2 functions as an evolutionarily conserved chaperone dedicated for uS5 [38,56,57].

Previous work had already suggested the involvement of the yeast homolog of PDCD2/PDCD2L in ribosome biogenesis. Specifically, a screen for candidate genes involved in ribosome biogenesis identified *TSR4* (Twenty S rRNA accumulation 4) as a gene required for 40S ribosomal subunit production [58]. A few years later, it was reported that Zfrp8 (Tsr4/PDCD2/PDCD2L homolog in *Drosophila*) depletion results in reduced cytoplasmic level of three RPs, including uS5 [59]. Consistent with these observations, PDCD2 copurifies with uS5 in both yeast and human cells and show binding via two-hybrid assays [38,56,57], suggesting a direct interaction between PDCD2 and uS5 that is evolutionarily conserved. Although the structure of the uS5–PDCD2 complex remains to be determined experimentally, we used AlphaFold-Multimer [60] to generate models of the human uS5–PDCD2 complex. Figure 3A shows the best confident relaxed structure with the highest predicted Local Distance Difference Test (pLDDT). Alternative predicted models showed highly similar pLDDT values, indicating uniformity among the predicted structures. Whereas the overall globular structure of uS5 remained largely unchanged in the context of the uS5–PDCD2 heterodimer relative to the uS5 monomer, residues 20–50 in the N-terminal disordered region of uS5 exhibited an increased confidence score and a considerably reduced predicted position error in the uS5–PDCD2 complex compared to the same region in the uS5 monomer (Figure 3B). In contrast, the C-terminal region of uS5 (aa 273–293) appears to be more disordered in the context of the uS5–PDCD2 complex relative to the uS5 monomer (Figure 3B). Interestingly, the disordered N-terminal extension of uS5 (see Figure 1B) is predicted to fold into a hydrophobic pocket located in the N-terminal half of human PDCD2 (Figure 3C). Notably, two phenylalanine residues of human uS5 (Phe25 and Phe29) are buried inside hydrophobic core regions of PDCD2 (Figure 3D). Consistent with this model, an FXXXFG motif can be found in the N-terminal extension of uS5 from humans, fruit flies, nematodes, and plants, whereas a single phenylalanine-glycine (FG motif) is found in uS5 from budding and fission yeasts (see Figure 1D). Although this remains a predicted model, the rearrangement of the uS5 unstructured N-terminal extension into a relatively stable structure in the context of the uS5–PDCD2 heterodimer is consistent with data in yeast showing that the first 42 amino acids of uS5 appear sufficient for interaction with *S. cerevisiae* Tsr4 [56,57]. The minimal PDCD2 interaction domain of uS5 in metazoans remains to be determined. The AlphaFold-Multimer prediction of the human uS5–PDCD2 complex also suggests the insertion of the uS5 dsRBD into a C-shaped opening formed by amino acids 204 to 239 of PDCD2 (Figure 3A,E), which is likely to stabilize the heterodimer.

Studies in both yeast and human cells indicate that Tsr4/PDCD2 recognizes uS5 co-translationally and that Tsr4/PDCD2 is required for the accumulation of newly synthesized uS5 [38,56,57]. Consistent with the view that Tsr4/PDCD2 recognizes nascent uS5 is the fact that the N-terminal disordered region of uS5 is required for the formation of a stable Tsr4–uS5 complex in yeast [56,57], which is also suggested by the prediction of the human PDCD2–uS5 complex shown in Figure 3. The underlying mechanism of the specific co-translational recruitment remains unclear, however. It is possible that PDCD2/Tsr4 might have some degree of affinity for the *uS5* mRNA, and thus, that the recruitment is initiated prior to *uS5* translation initiation. The loss of function of PDCD2/Tsr4 phenocopies that of uS5 deficiency: reduced 40S production; 20S and 21S pre-rRNA accumulation in yeast and humans, respectively; and reduced incorporation of uS5 into pre-40S particles [38,56,57]. These findings, the co-translational binding of PDCD2 to nascent uS5, and the lack of identification of ribosome assembly factors in the interaction network PDCD2 support a conserved role of PDCD2 as a dedicated chaperone to uS5 [38,56,57].

How does PDCD2 promote the incorporation of uS5 into the pre-40S particle? In human cells, the model (Figure 4, see steps 1–4) suggests that once the complex is formed co-translationally, PDCD2 escorts uS5 until it is incorporated into pre-40S particles in the nucleolus. In support of this model, the PDCD2–uS5 complex can be found in the cytoplasm, the nucleus, and the nucleolus in human cells [38]. Furthermore, upon depletion of PDCD2, reduced levels of uS5 are detected in the nucleolus [38]. Interestingly, the chaperoning function of Tsr4 appears to be restricted to the cytoplasm in *S. cerevisiae* [56], suggesting some differences in the mechanism of action of yeast Tsr4 and human PDCD2. Despite the essential role of PDCD2/Tsr4 in chaperoning nascent uS5, the mechanism of uS5 nuclear import and whether PDCD2/Tsr4 is required for this process remain unclear. Yet, recent work from the Pertschy lab (Brigitte Pertschy, personal communication) suggests the presence of two independent nuclear localization signals (NLS) in *S. cerevisiae* uS5, one located in the arginine-rich region between residues 10 through 28, which overlap with the Tsr4 binding site, and a second NLS between residues 76 through 145 that interacts with the importin Pse1 (human IPO5). Although we now have a reasonable understanding of how PDCD2/Tsr4 promotes uS5 incorporation into pre-40S particles by forming a stable complex with uS5, the mechanism by which this complex is disassembled remains to be determined. As structures of the eukaryotic ribosome show a connection between the N- and C-terminal region of uS5 [14,61], Black et al. [56] proposed the possibility that an intramolecular interaction between the N- and C-terminal regions of uS5 could destabilize the interaction between the N-terminal region of uS5 and PDCD2/Tsr4.

## 6. Conserved Role of PDCD2 in Stem Cell Biology and Embryonic Development

As mentioned previously, several studies using different multicellular model organisms report important roles for PDCD2 and homologs in stem cells and embryonic development. Zfrp8, the ortholog of PDCD2 in fruit flies, is important for the maintenance of two types of stem cells found in the *Drosophila* ovary: germ stem cells (GSC) and follicle stem cells (FSC). When Zfrp8 function is lost, GSCs and FSCs stop dividing, while differentiated cells show no growth phenotype [62]. Hematopoietic stem cells (HSC) are also greatly affected by the loss of Zfrp8 in *Drosophila*, as a Zfrp8 deficiency impedes self-renewal of HSCs but has no effect on pluripotent precursors [63]. In humans, PDCD2 was shown to be important for hematopoietic stem/progenitor cells viability and essential for erythroid differentiation and development [64]. Hematopoiesis is also impaired by the knockdown of PDCD2 in zebrafish embryos. Specifically, loss of *pdcd2* expression prevents HSC emergence/initiation and maintenance, in addition to causing erythroid differentiation arrest [65]. In human cells and zebrafish, ineffective hematopoiesis resulting from PDCD2 deficiency is associated with cell cycle defects [64,65]. As mentioned earlier, embryonic stem cells’ viability and proliferation are dependent on PDCD2 in mice. A *PDCD2* deletion leads to embryonic development defects, with fertilized eggs attaining morula or blastocyst stages but not developing further [55]. *zfrp8*-null *Drosophila* also show abnormal development phenotypes consisting of developmental delay and lethality at larval stages [66]. In the silkworm (*Bombyx mori*), it was recently shown that BmZfrp8, the PDCD2 homolog, is expressed at different days, with a peak of expression in the middle of embryonic development [67], and this led the authors to suggest that BmZfrp8 is essential for the regulation of growth and development. Collectively, the aforementioned studies underline the importance of PDCD2 and its orthologs in stem cells’ survival and embryonic development. More studies are therefore needed to elucidate the molecular mechanisms underlying the critical role of PDCD2 in stem cell biology and development and whether this role depends on the function of PDCD2 as a dedicated RP chaperone.

## 7. PDCD2L: A Paralog of Human PDCD2 That Associates with uS5

As will be discussed below, the ancestral duplication of the *PDCD2L* gene appears to be beneficial to organisms, as PDCD2L and PDCD2 paralogs participate in complementary functions involved in ribosome biogenesis in human cells [39]. As for PDCD2, PDCD2L physically associates with uS5. Specifically, affinity purification of GFP-PDCD2L from human cells coupled with mass spectrometry (AP-MS) revealed uS5 to be a strong interacting protein [39], and, reciprocally, PDCD2L copurifies with uS5 [68]. Interestingly, PDCD2 and PDCD2L show a mutually exclusive interaction with uS5, as the AP-MS assays of PDCD2L do not identify PDCD2 as a binding partner and vice versa [39]. However, and in contrast to PDCD2, the analysis of PDCD2L interactions revealed its association with several late 40S maturation factors [39]. Furthermore, one of the last precursors of the mature 18S rRNA, namely the 18S-E pre-rRNA, also specifically copurifies with PDCD2L, strongly suggesting that PDCD2L associates with late pre-40S particles [39]. Interestingly, analysis of the PDCD2L amino acid sequence revealed the presence of a leucine-rich nuclear export signal (NES), which was confirmed to act as a functional NES since (i) PDCD2L associates with XPO1/CRM1 and (ii) mutations in the PDCD2L NES prevents its association with XPO1/CRM1 and cause the accumulation of PDCD2L in the nucleus [39]. Notably, human cells deficient in PDCD2L show a marked accumulation of free 60S ribosomal subunits, a hallmark of 40S subunit deficiency. The absence of PDCD2L alone does not affect the maturation of 18S rRNA precursors, however, but clearly makes human cells more sensitive to the depletion of PDCD2, as the absence of both PDCD2 and PDCD2L exacerbates the ribosome biogenesis defects associated with the single PDCD2 depletion [39]. Together, the current data suggest that PDCD2L could act as a protein adapter for XPO1/CRM1 in the nuclear export of the pre-40S subunit. Yet, because PDCD2L is not essential for the export of 40S subunit precursors [39], it likely shares that function with one or several other ribosome maturation factors.

In contrast to the *PDCD2* and *TSR4* genes, *PDCD2L* is not essential for cell viability. As shown by the cancer dependency map, most cancer cell lines survive a deletion of *PDCD2L* [35]. However, PDCD2L seems to be required for embryonic development in mice, as *Pdcd2l*-null embryos were resorbed at mid and late gestation and no homozygous offsprings were born from heterozygous breeding [69]. In agreement with these results, knockdown of *trus*, the ortholog of *PDCD2L* in *Drosophila melanogaster*, causes a high rate of lethality at the third instar larval stage [69]. These results therefore suggest an important role for PDCD2L during embryogenesis. Further studies will be necessary in order to determine whether the role of PDCD2L during development depends on its association with uS5 and its function in ribosome biogenesis.

In summation, the available data indicate that the sequential interaction of PDCD2 and PDCD2L with uS5 contributes to different steps in 40S ribosomal subunit production, which is consistent with the additive ribosome biogenesis defects caused by the double depletion of PDCD2 and PDCD2L compared to single depletions [39]. Accordingly, as shown in Figure 4, we propose a working model of how PDCD2 and PDCD2L function in ribosome biogenesis via their mutually exclusive association with uS5. Specifically, published work supports a model (Figure 4) wherein human PDCD2 functions as a dedicated chaperone by recruiting uS5 co-translationally and by facilitating its incorporation into nucleolar pre-40S particles. Subsequently, PDCD2L would associate with nucleolar pre-40S particles via binding to uS5 and contribute to the efficient nuclear export of 40S precursors.

## 8. uS5 Arginine Methylation and uS5–PRMT3 Complex

An outstanding question in the field of ribosome function and regulation is the biological role of RP post-translational modifications (PTMs). Indeed, whereas RPs are subject to a variety of PTMs [70,71,72,73], few RP-modifying enzymes have been identified and studied to date. Arginine is the predominant methylated amino acid in both the eukaryotic 40S and 60S ribosomal subunits [74]. Methylation of RPs at arginines is evolutionarily conserved [75,76,77] and fluctuates during the cell cycle [78]. Although the methylation of arginine residues is not expected to alter the net charge of RPs, it can, however, change protein hydrophobicity and influence interactions with proteins and nucleic acids, thereby affecting functional properties such as stability, subcellular localization, complex assembly/disassembly, etc. [79]. Yet, the functional roles of RP methylation remain poorly understood. Arginine methylation is catalyzed by protein arginine methyltransferases (PRMTs), an evolutionarily conserved family of enzymes divided into two major classes depending on the type of dimethylarginine they generate: type I PRMTs modify proteins through the catalysis of asymmetric dimethylarginine, whereas type II PRMTs catalyze the formation of symmetric dimethylarginine [79,80]. Studies in the fission yeast *S. pombe* have led to the identification of PRMT3 as the first eukaryotic RP methyltransferase via arginine methylation of uS5 [17]. uS5 methylation by PRMT3 was subsequently demonstrated in human cells and in mice [19,81]. Although PRMT3 has no homolog in *S. cerevisiae* [80], budding yeast uS5 is arginine methylated by Hmt1 (homolog of human PRMT1) [18], and uS5 arginine methylation levels were found to increase during the stationary phase of *S. cerevisiae* [82]. In all of these species, uS5 was shown to be methylated on arginine residues located in its N-terminal RG-rich region [18,19,83].

PRMT3 is a primarily cytosolic type I arginine methyltransferase which possesses, in addition to its methyltransferase domain, a C2H2 zinc finger domain [84]. Although the zinc finger domain of PRMT3 is not required for methylation of an artificial substrate in vitro, it is necessary for the recognition of substrates in cell extracts [85]. Accordingly, the zinc finger domain of PRMT3 is necessary for binding uS5 in yeast and human cells [19,83]. As the structure of the PRMT3–uS5 complex has not yet been determined, we used AlphaFold-Multimer to visualize a predicted model of the human complex (Figure 5A). Here again, the best confident relaxed structure with the highest pLDDT score is presented, and the alternative predicted models showed highly similar pLDDT values. Surprisingly, direct physical contacts between the zinc finger domain of PRMT3 and residues of uS5 are not predicted in the model of the PRMT3–uS5 complex (Figure 5B), suggesting that the single zinc finger domain of PRMT3 is critical to fold its N-terminus into a structure that stably recognizes uS5. With regard to uS5 methylation by PRMT3, the model nicely predicts the arrangement of the uS5 RG-rich region proximal to the catalytic center of PRMT3 (Figure 5C) with a conserved glutamic acid residue involved in catalysis [83] located near arginine residues known to be methylated in uS5 (Figure 5D). Although arginine-methylated versions of uS5 appear to be part of actively translating ribosomes [81], it remains unclear when and where uS5 gets methylated by PRMT3: before or after incorporation into pre-40S ribosomal subunits. Whereas *S. pombe* and human uS5 form a complex with PRMT3 that is sufficiently stable to be easily isolated by affinity purification [17,19,39,83], only a small proportion of PRMT3 appears to co-sediment with the free 40S subunit [17,19].

The biological significance of uS5 arginine methylation by PRMT3 remains poorly understood. In the past decade, there has been increasing interest in the idea that decoration of RPs with various modifications could customize ribosomes to translate a subset of functionally related mRNAs [86,87]. Our data in yeast and human cells show that the absence of uS5 methylation in *PRMT3* knockout cells results in a global shift in uS5 migration as analyzed by SDS-PAGE (Figure 5E, compare lanes 2–4 to lane 1) [83]. Although these results do not exclude the idea that uS5 methylation could contribute to ribosome heterogeneity, they suggest that the large majority of uS5 gets arginine methylated by PRMT3. Several studies have investigated the functional role of uS5 methylation via genetic alteration of *PRMT3* in various model organisms. In *S. pombe*, the deletion of *rmt3* results in increased levels of free 60S subunits, although pre-rRNAs processing is not disturbed [17]. Interestingly, the imbalance in free ribosomal subunits in *rmt3*-null *S. pombe* is not the consequence of deficient uS5 methylation but is rather due to the absence of uS5–Rmt3 interaction, as a methyltransferase-dead version of Rmt3 rescues the increased levels of free 60S subunits observed in *rmt3*-null cells [83]. Consistent with the view that the PRMT3–uS5 interaction is functionally important, studies in human cells indicate that PRMT3 stabilizes uS5 by inhibiting its ubiquitination and degradation by the proteasome [88]. Conversely, siRNA-mediated depletion of uS5 in humans cells considerably reduces the total cellular abundance of PRMT3 [68], suggesting that the uS5–PRMT3 interaction reciprocally stabilizes PRMT3. In mice, a targeted insertion in intron 14 of *PRMT3* that is predicted to remove the last 34 amino acids results in embryos that are smaller in size, but intriguingly, this size difference is lost by weaning age [81]. In contrast to the imbalance in free ribosomal subunits observed in *rmt3*-deleted fission yeast cells, mouse embryonic fibroblasts cultured from *PRMT3* mutant embryos showed a normal level of free ribosomal subunits, 80S monosomes, and polysomes [81]. The impact of the deletion of the *PRMT3* homolog was also studied in the plant *Arabidopsis thaliana*. In this species, the absence of AtPRMT3 alters the polysome profile and affects ribosome biogenesis [89]. More recently, the same group reported that the function of AtPRMT3 in ribosome biogenesis is primarily mediated by the physical interaction with RPS2B (one of four proteins encoded by genes orthologs of uS5 in *Arabidopsis*), but independent of its methyltransferase activity [90], consistent with previous findings using fission yeast [83]. Although the underlying mechanism by which the AtPRMT3–RPS2B interaction contributes to ribosome biogenesis remains to be defined, the authors propose that AtPRMT3 acts as a chaperone for RPS2B, preventing non-specific interactions of RPS2B and promoting its incorporation into pre-ribosomes [90].

## 9. ZNF277: The Newest Member among Conserved uS5-Associated Proteins

The discovery of PDCD2 and PDCD2L as novel uS5-associated proteins [38,39] beyond PRMT3 stimulated a comprehensive analysis of the human uS5 interactome. In addition to PRMT3, PDCD2, PDCD2L, RPs, and 40S maturation factors (MFs), we have recently identified a poorly characterized zinc finger protein, ZNF277, among the top 10% of uS5-associated proteins [68]. Importantly, the *uS5* mRNA is specifically enriched in ZNF277 precipitates [68], suggesting that ZNF277 is recruited co-translationally by nascent uS5, a frequent feature of dedicated RP chaperones [91,92]. A complex between ZNF277 and uS5 is also supported by independent studies that used high-throughput affinity purifications coupled with mass spectrometry in human cells [93] as well as analysis in *Drosophila* [94] and *C. elegans* [95]. These findings therefore support the existence of an evolutionarily conserved physical connection between ZNF277 and uS5.

Human ZNF277 contains five C2H2 zinc finger motifs, two featuring the typical amino acid consensus sequence (C-x(2,4)-C-x(3)-[LIVMFYWC]-x(8)-H-x(3,5)-H) and three containing atypical consensus motifs. Similar to PRMT3, the interaction between uS5 and ZNF277 depends on the integrity of its zinc finger domains, especially the two most C-terminal zinc fingers of ZNF277 [68]. Furthermore, current data support the view that ZNF277 and PRMT3 compete for uS5 binding: overexpression of wild-type PRMT3 in human cells inhibited the formation of the ZNF277–uS5 complex, whereas knockdown of ZNF277 resulted in increased levels of uS5 in PRMT3 precipitates [68]. These results therefore suggest that PRMT3 and ZNF277 have a common binding site on uS5. Although current proteomics data indicate that the PRMT3–uS5 complex is more abundant compared to the ZNF277–uS5 complex in human embryonic kidney cells [38,68], this stoichiometry may be different in other cell types.

To date, the molecular and cellular function of ZNF277 remains unclear. The homolog of human ZNF277 in mice, Zfp277, was shown to function as a transcriptional regulator [96] and to impact cellular proliferation and senescence [97]. In human cells, the depletion of ZNF277 does not appear to affect ribosome profiles as determined by polysome assays, despite direct physical interaction with uS5 and the localization of a fraction of ZNF277 to nucleoli in human cells [68]. Recent work in the nematode *C. elegans* shows that the homolog of human ZNF277, ZTF-7, as well as 40S RPs are required for the nucleolar depletion of the RNA exosome after a cold shock [95]. As the RNA exosome is a key complex required for pre-rRNA processing [7], these results suggest that ZNF277 may be involved in the regulation of ribosome biogenesis. It is also interesting to note that, similarly to PRMT3 conservation, *S. cerevisiae* does not appear to code for a protein with homology to ZNF277, whereas a ZNF277 homolog is found in the *S. pombe* genome.

## 10. Conclusions and Outlook

Since the initial discovery of the first non-ribosomal uS5-associated protein in fission yeast almost twenty years ago [17], studies have now identified a set of four evolutionarily conserved uS5-interacting partners: PDCD2, PDCD2L, PRMT3, and ZNF277. In this review, we highlighted the complex and conserved regulatory network responsible for monitoring the availability and the folding of uS5 for the formation of 40S ribosomal subunits. To our knowledge, uS5 is the RP with the greatest extent of conserved associated proteins outside of the ribosome. While PDCD2 and PDCD2L (and their homologs) contribute to the function of uS5 inside the ribosome, the functional significance of the PRMT3–uS5 (as well as uS5 arginine methylation by PRMT3) and ZNF277–uS5 complexes remains to be established. As *ZNF277* overexpression is associated with improved prognosis of human cancers according to the Human Protein Atlas Project [98], whereas *PRMT3* overexpression appears to be associated with poor prognosis, uncovering the process by which ZNF277 and PRMT3, two C2H2-type zinc finger proteins, compete for uS5 binding is likely to have relevance to cancer biology. As the PRMT3–uS5 complex is exclusively cytosolic [68], why would cells benefit from retaining a fraction of uS5 in the cytoplasm? Could this allow for the repair of damaged ribosomes, exchange between methylated and unmethylated uS5 in 40S subunits, or produce uS5-deficient ribosomes, as was recently shown for eS26 in yeast [99]?

Further studies will also be required to clarify the role of uS5 in the nuclear export of pre-40S particles [9,11,32] and whether PDCD2L functions as an adaptor protein for the CRM1-mediated export of pre-40S subunits [39] in specific cell types, thereby explaining its critical role during embryonic development [69]. Establishing the functional contribution of PDCD2 in stem cell biology and embryonic development and whether the critical role of PDCD2 in these processes is linked to its role in ribosome biogenesis via uS5 will also be very interesting. Ultimately, the ecosystem of uS5-associated proteins is complicated by the fact that these five proteins likely also form different trimers with uS5 acting as a bridging protein: PRMT3–uS5–PDCD2, PRMT3–uS5–PDCD2L, ZNF277–uS5–PDCD2, and ZNF277–uS5–PDCD2L [68]. Further research will therefore be essential to understand how these complexes coexist, cooperate, or antagonize each other.

## Figures and Tables

**Figure 1 biomolecules-13-00853-f001:**
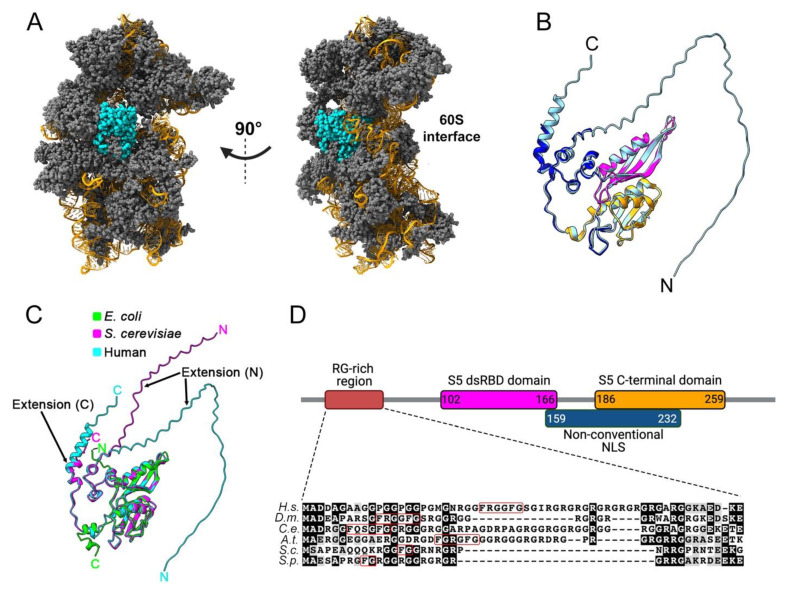
Structural characteristics of the 40S ribosomal protein uS5. (**A**) Cryo-EM structure of the actively translating 40 ribosomal subunit (PDB entry 5AJ0), left, and rotated 90 degrees, right. uS5 is shown in cyan, while the other 40S ribosomal proteins are colored in grey. The 18S rRNA is shown in orange. (**B**) Superposition of the tertiary structures of uS5 extracted from the active 40S ribosomal subunit (PDB entry 5AJ0; dark blue, magenta, and orange colors) and predicted by AlphaFold (pale blue). Note that the uS5 structure from the 40S subunit (PDB 5AJ0) represents only amino acids D57 to T278. The double-stranded RNA-binding-like domain and the conserved S5 C-terminal domain are shown in pink and orange, respectively, while the N- and C-terminal extensions are only seen in the AlphaFold model (pale blue). (**C**) Superposition of the AlphaFold tertiary structures of uS5 from *E. coli* (lime, P0A7W1), *S. cerevisiae* (magenta, P25443), and human (cyan, P15880) showing eukaryotic-specific N- and C-terminal extensions. (**D**) Motifs and functional domains of uS5 are shown. Numbers indicate the amino acid positions of each domain. Alignment and shading were generated using ClustalW and Boxshade software. Sequences are from *Homo sapiens* (H.s.), *Drosophila melanogaster* (D.m.), *Caenorhabditis elegans* (C.e.). *Arabidopsis thaliana* (A.t.), *Saccharomyces cerevisiae* (S.c.), and *Schizosaccharomyces pombe* (S.p.). The FXXXFG and FG motifs are boxed in red.

**Figure 2 biomolecules-13-00853-f002:**
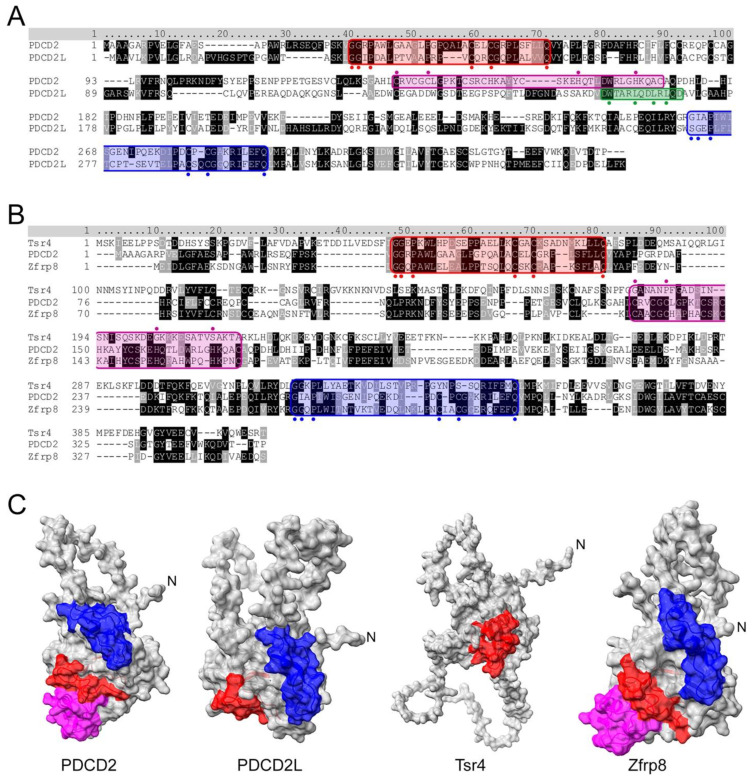
Sequence and structural analysis of PDCD2 and PDCD2L paralogs. (**A**) Amino acid sequence alignment of human PDCD2 and PDCD2L. Both proteins harbor N- and C-terminal TYPP domains (highlighted in red and blue, respectively) with conserved GxxP, Cx_1-2_C, and Q residues highlighted with circles. Whereas PDCD2 contains a MYND zinc finger domain (in magenta with critical cysteine and histidine residues indicated by circles marked above), PDCD2L harbors a leucine-rich NES consensus sequence (green), Φx_2-3_Φx_2-3_ΦxΦ, where Φ represents large hydrophobic residues (indicated by green circles marked underneath). (**B**) *S. cerevisiae* Tsr4 lacks the MYND zinc finger domain and its C-terminal TYPP domain is degenerated. (**C**) AlphaFold structures for human PDCD2 (Q16342), human PDCD2L (Q9BRP1), yeast Tsr4 (P25040), and *Drosophila* Zfrp8 (Q9W1A3). Red: N-terminal TYPP domain; Blue: C-terminal TYPP domain; Magenta: MYND domain.

**Figure 3 biomolecules-13-00853-f003:**
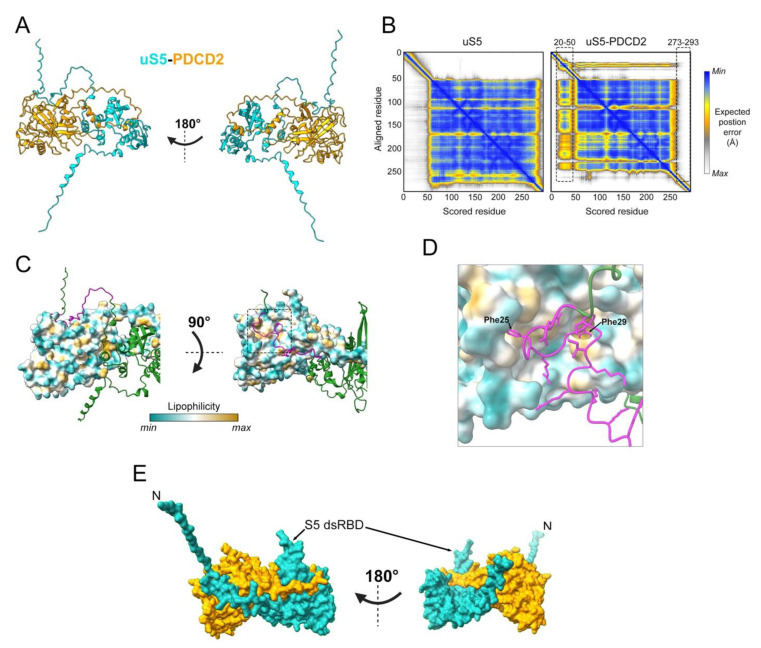
Predicted model of the human uS5–PDCD2 complex. (**A**) AlphaFold-Multimer [60] prediction of the human uS5 (cyan)–PDCD2 (orange) complex, left, and rotated 180 degrees, right. (**B**) AlphaFold-predicted aligned error plot for the uS5 monomer (left) and uS5–PDCD2 complex (right), highlighting residues 20–50 of uS5 confidently predicted to interact with PDCD2 and residues 273–293 that show reduced predicted position error. (**C**) Surface representation of PDCD2 lipophilicity with ribbon-like structure of uS5 (green), left, and rotated 90 degrees, right. Residues 20–50 of uS5 are colored in magenta. (**D**) Phe25 and Phe29 residues of human uS5 are predicted to be embedded in hydrophobic core regions of PDCD2. (**E**) Surface representation of the uS5 (cyan)–PDCD2 (orange) complex, left, and rotated 180 degrees, right. A C-shaped region of PDCD2 (aa 204–239) wraps around the S5 dsRBD of uS5.

**Figure 4 biomolecules-13-00853-f004:**
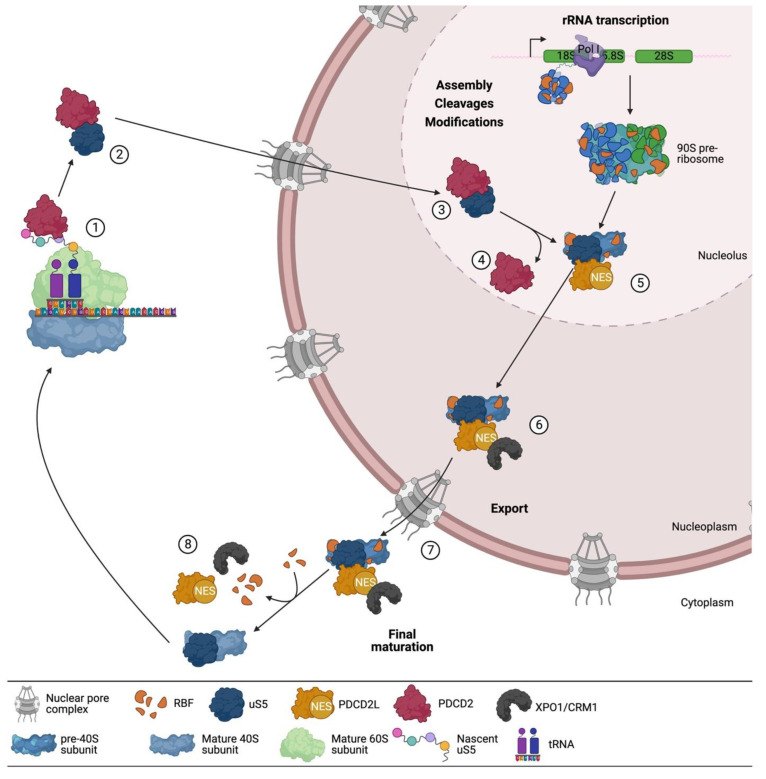
Model of how human PDCD2 and PDCD2L contribute to 40S ribosomal subunit biogenesis via interaction with uS5. (1) PDCD2 binds nascent uS5 co-translationally. (2) Interaction between uS5 and PDCD2 takes place in the cytoplasm and the (3) nucleolus. The role of PDCD2 in the nuclear import of uS5 remains to be determined. (4) In the nucleolus, PDCD2 would promote the incorporation of uS5 into the pre-40S ribosomal subunit. (5) PDCD2L binds to pre-40S subunits in the nucleolus via interaction with uS5. (6) The leucine-rich NES of PDCD2L promotes the recruitment of XPO1/CMR1 to the pre-40S particles. (7) Once pre-40S particles are exported to the cytoplasm, (8) PDCD2L and XPO1/CRM1 would dissocociate from 40S precursors.

**Figure 5 biomolecules-13-00853-f005:**
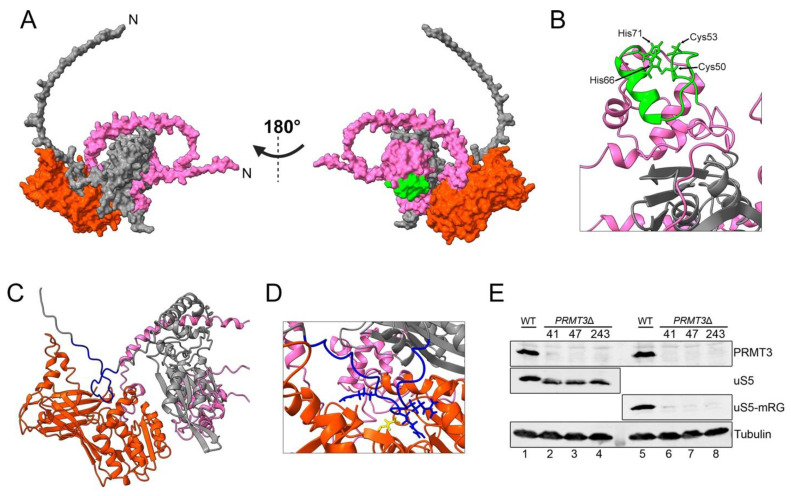
Predicted model of the human uS5–PRMT3 complex.(**A**) AlphaFold-Multimer [60] prediction of the human uS5 (grey)–PRMT3 (magenta) complex, left, and rotated 180 degrees, right. The C2H2 zinc finger (aa 48–71) and methyltransferase (aa 217–531) domains are shown in green and red, respectively. (**B**) The zinc finger domain of PRMT3 does not contact uS5. Shown is the zinc finger (green) of PRMT3 (magenta) with critical cysteine and histidine residues. Human uS5 is shown in grey. (**C**) The RG-rich region of human uS5 (aa 34–52, shown in dark blue) is located proximally to the catalytic center of the methyltransferase domain (red) of PRMT3. (**D**) Internal view of the PRMT3 methyltransferase domain with the Glu-338 critical for catalysis shown in yellow and arginine residues 42, 44, and 46 of uS5 shown in dark blue. (**E**) Western blot analysis using total extracts prepared from three independent clonal lines of HeLa cells deleted for *PRMT3* (lanes 2–4 and 6–8) and wild-type (lanes 1 and 5) HeLa cells. Lanes 1–4 were analyzed for total uS5, while lanes 5–8 were analyzed for arginine-methylated uS5 (uS5-mRG).

## Data Availability

Not applicable.

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
