# Peer review of "Ribosomal Protein uS5 and Friends: Protein–Protein Interactions Involved in Ribosome Assembly and Beyond"

_biomolecules, 2023, doi:10.3390/biom13050853_

Round 1

Reviewer 1 Report

The review by Landry-Voyer and coauthors focuses on the network of interactions between ribosomal protein Rps2/uS5 and four extraribosomal factors, PDCD2, PDCD2L, PRMT3, and ZNF277. The existence of extraribosomal complexes of ribosomal proteins is a poorly understood phenomenon. The relatively recent experimental demonstration that ribosomal proteins require cotranslational association with dedicated chaperones for stabilization and transport to the site of ribosome assembly has advanced our understanding of why some of these complexes exist. The Bachand group published several manuscripts in the past on the role of Rps2/uS5-associated proteins, and in 2020 they showed that PDCD2 functions as a dedicated uS5 chaperone. This review summarizes data from their previous work on these proteins in the context of current literature and suggests several ideas for further exploration.

The manuscript begins logically with uS5, providing a brief overview of the cryo-EM data on the position of uS5 in the structure of the SSU, the function of the protein in mature ribosomes, and the requirement of uS5 for SSU assembly. Although the role of uS5 in translation is not the main focus of the review, the authors may consider additionally mentioning here studies from Erik Böttger’s group, which has published several papers on the biological effects of ram mutations in the mouse, including the A226Y mutation in Rps2 (e.g., doi:10.1038/s42003-021-02204-z).

The authors then continue with the structural description of PDCD2/PDCD2L paralogs and their counterparts in yeast (Tsr4) and Drosophila (Zfrp8). The structural modeling of the uS5-PDCD2 complexes via AlphaFold-Multimer (Fig.3) provides an interesting addition to the previously published data, notably as it predicts specific amino acids at key positions of the complexes. While a speculative exercise at this point, it certainly provides candidates for future experimental validation. Overall, this part provides a clearly written summary of what is known about the chaperone role of these uS5-interacting proteins (mainly from the mammalian and yeast systems) and where to go next.

The section regarding the effects of the loss of function of PDCD2 orthologs on development covers the body of the literature that is still largely disconnected from this protein’s function as the uS5 chaperone. Likewise, more studies are needed to elucidate the role of PDCD2L. Perhaps in Fig. 4, the authors should make it more clear that the parts of the model regarding PDCD2L are hypothetical at this point, whereas the PDCD2 data are more solid; this may not be not obvious to a reader who starts with the figure rather than carefully reading the text.

The last two members of the uS5-binding network are the arginine methyltransferase PRMT3 and zinc-finger protein ZNF277, the molecular and cellular functions of which are not yet fully understood.  The AlphaFold modeling of the uS5-PRMT3 complex again makes interesting predictions, such as the lack of interaction between uS5 and the zinc finger in PRMT3. The ZNF277 section is mainly descriptive.

Overall, the manuscript provides a balanced review of literature, spanning 1969 to 2023 and includes key references included as far as I can tell. The authors’ citation of their own work is unavoidable in the context of this review since, historically, their group provided key insights into the functions of several of these proteins, and I do not see it as a problem. In this reviewer’s opinion, the review is comprehensive and clearly written. It brings together a lot of useful information on the ribosomal protein Rps2/uS5, its chaperones, and interacting proteins.The data from previous studies are augmented by AlphaFold-based modeling of these protein complexes; these models, albeit hypothetical, will stimulate further experimentation and, as such, I think they are appropriate in this review. The authors have made important contributions to studies of some of these proteins. Their expert opinion on these factors is a valuable contribution to the field that creates a unique resource and does not duplicate any other published review, to the best of my knowledge. I support the publication of this review article.

Two small typos that I noticed:

Line 248 – glycine

Line 384 – should be ‘facilitating’?

Author Response

The review by Landry-Voyer and coauthors focuses on the network of interactions between ribosomal protein Rps2/uS5 and four extraribosomal factors, PDCD2, PDCD2L, PRMT3, and ZNF277. The existence of extraribosomal complexes of ribosomal proteins is a poorly understood phenomenon. The relatively recent experimental demonstration that ribosomal proteins require cotranslational association with dedicated chaperones for stabilization and transport to the site of ribosome assembly has advanced our understanding of why some of these complexes exist. The Bachand group published several manuscripts in the past on the role of Rps2/uS5-associated proteins, and in 2020 they showed that PDCD2 functions as a dedicated uS5 chaperone. This review summarizes data from their previous work on these proteins in the context of current literature and suggests several ideas for further exploration.

RESPONSE: We thank the Reviewer for the positive assessment.

The manuscript begins logically with uS5, providing a brief overview of the cryo-EM data on the position of uS5 in the structure of the SSU, the function of the protein in mature ribosomes, and the requirement of uS5 for SSU assembly. Although the role of uS5 in translation is not the main focus of the review, the authors may consider additionally mentioning here studies from Erik Böttger’s group, which has published several papers on the biological effects of ram mutations in the mouse, including the A226Y mutation in Rps2 (e.g., doi:10.1038/s42003-021-02204-z).

RESPONSE: Excellent suggestion. We were not aware of these studies from the Böttger’s lab. Accordingly, the revised manuscript mentions the effects of the uS5 A226Y ram mutation in human cells and mice. This is mentioned on lines 139-142 of the revised manuscript. We thank the reviewer for alerting us of these studies.

The authors then continue with the structural description of PDCD2/PDCD2L paralogs and their counterparts in yeast (Tsr4) and Drosophila (Zfrp8). The structural modeling of the uS5-PDCD2 complexes via AlphaFold-Multimer (Fig.3) provides an interesting addition to the previously published data, notably as it predicts specific amino acids at key positions of the complexes. While a speculative exercise at this point, it certainly provides candidates for future experimental validation. Overall, this part provides a clearly written summary of what is known about the chaperone role of these uS5-interacting proteins (mainly from the mammalian and yeast systems) and where to go next.

RESPONSE: We thank the Reviewer for the positive assessment.

The section regarding the effects of the loss of function of PDCD2 orthologs on development covers the body of the literature that is still largely disconnected from this protein’s function as the uS5 chaperone. Likewise, more studies are needed to elucidate the role of PDCD2L. Perhaps in Fig. 4, the authors should make it more clear that the parts of the model regarding PDCD2L are hypothetical at this point, whereas the PDCD2 data are more solid; this may not be not obvious to a reader who starts with the figure rather than carefully reading the text.

RESPONSE: We thank the Reviewer for the positive assessment. Although we respect the reviewer’s opinion, we do not feel that the data on PDCD2 are more solid than on PDCD2L. The fact that there are likely several nuclear export adaptors for pre-40S subunits in addition to PDCD2L, whereas PDCD2 is likely to only uS5 dedicated chaperone, results in a more explicit phenotype after a PDCD2 loss-of-function compared to a PDCD2L loss-of-function. We think that this is clearly explained in lines 402-404 of the manuscript and does not required clarification in the legend to Figure 4.

The last two members of the uS5-binding network are the arginine methyltransferase PRMT3 and zinc-finger protein ZNF277, the molecular and cellular functions of which are not yet fully understood.  The AlphaFold modeling of the uS5-PRMT3 complex again makes interesting predictions, such as the lack of interaction between uS5 and the zinc finger in PRMT3. The ZNF277 section is mainly descriptive.

RESPONSE: We thank the Reviewer for the positive assessment.

Overall, the manuscript provides a balanced review of literature, spanning 1969 to 2023 and includes key references included as far as I can tell. The authors’ citation of their own work is unavoidable in the context of this review since, historically, their group provided key insights into the functions of several of these proteins, and I do not see it as a problem. In this reviewer’s opinion, the review is comprehensive and clearly written. It brings together a lot of useful information on the ribosomal protein Rps2/uS5, its chaperones, and interacting proteins.The data from previous studies are augmented by AlphaFold-based modeling of these protein complexes; these models, albeit hypothetical, will stimulate further experimentation and, as such, I think they are appropriate in this review. The authors have made important contributions to studies of some of these proteins. Their expert opinion on these factors is a valuable contribution to the field that creates a unique resource and does not duplicate any other published review, to the best of my knowledge. I support the publication of this review article.

RESPONSE: We thank the Reviewer for the positive assessment.

Two small typos that I noticed:

Line 248 – glycine

RESPONSE: Corrected (now line 278).  Thank you.

Line 384 – should be ‘facilitating’?

RESPONSE: Corrected (now line 422).  Thank you.

Reviewer 2 Report

This is a nice and important review. It describes the potential roles of uS5 interaction proteins, PRMT3, PDCD2, PDCD2L, and ZNF277, highlighting the complex and conserved regulatory network responsible for monitoring the availability and the folding of uS5. There are only a few comments below.

1.      It would be nice to see the structural differences among bacterial, yeast, and human uS5. The readers will know better about the EES (eukaryotic extension sequence) of uS5.

2.      Line 96-97: Interestingly, several arginine residues in the N-terminal RG-rich extension of uS5 are targeted by asymmetric demethylation…

The study below indicates that the methylation near the NLS alters the transporter interaction. Is it possible PRMT3 also modulates the import of uS5 like this? Please include the discussion about the potential role of methylation of ribosomal proiteins.

 Nonclassical nuclear localization signals mediate nuclear import of CIRBP (PNAS, 2020, 117 (15) 8503-8514, https://doi.org/10.1073/pnas.1918944117)

3.      Line 159-164: please add the citations related to the description.

4.      Line 163-164: It would be nice to see these three proteins, uS2-uS5-eS21, on the pre-40S. This will let the readers understand the relative positions and interactions and know how they are loaded as a cluster.

5.      The color labeling in Fig 2 should be consistent in A and C, making it easier for the readers to compare.

(A) N- and C-terminal TYPP 204 domains (highlighted in red and blue, respectively)

(C) Blue: N-terminal TYPP domain, Red: C-terminal TYPP domain

6. The data in Fig 3 and Fig   5 are generated by AlphaFold-Multimer in this manuscript. More than one prediction model should be given from the program. Thus, please indicate the detailed information and factors of the selective structure. 

7. Fig 5A: The label of the figure should be changed to 180-degree.

Author Response

REVIEWER #2

This is a nice and important review. It describes the potential roles of uS5 interaction proteins, PRMT3, PDCD2, PDCD2L, and ZNF277, highlighting the complex and conserved regulatory network responsible for monitoring the availability and the folding of uS5. There are only a few comments below.

RESPONSE: We thank the Reviewer for the positive assessment.

  1. It would be nice to see the structural differences among bacterial, yeast, and human uS5. The readers will know better about the EES (eukaryotic extension sequence) of uS5.

RESPONSE: Good point. As suggested by reviewer #2, we have now added superimposed models of uS5 from E. coli, S. cerevisiae, and humans to highlight the eukaryotic-specific N- and C-terminal uS5 extensions. The new figure is presented as Fig. 1C and described on lines 93-95 of the revised manuscript.

  1. Line 96-97: Interestingly, several arginine residues in the N-terminal RG-rich extension of uS5 are targeted by asymmetric demethylation… The study below indicates that the methylation near the NLS alters the transporter interaction. Is it possible PRMT3 also modulates the import of uS5 like this? Please include the discussion about the potential role of methylation of ribosomal proiteins. Nonclassical nuclear localization signals mediate nuclear import of CIRBP (PNAS, 2020, 117 (15) 8503-8514, https://doi.org/10.1073/pnas.1918944117)

RESPONSE: Our review includes a section on arginine methylation of uS5 and ribosomal protein in general (lines 425-518). As discussed in our review, arginine methylation is frequent in eukaryotic ribosomal proteins; yet, the functional roles of ribosomal protein methylation remain poorly understood. As for CIRBP, methylation of arginine residues near an NLS in a ribosomal protein could potentially affect its subcellular localization, but can also affect many other biochemical properties of proteins, including stability, binding to DNA and RNA, hydrophobicity, etc. As suggested by the review, we have added such a discussion in the revised version of our review (see line 431-435).

  1. Line 159-164: please add the citations related to the description.

RESPONSE:  The citations describing the entry state of uS5 in yeast and human ribosomes was added to the revised manuscript (line 173-179).  We thank the reviewer for noting this oversight.

  1. Line 163-164: It would be nice to see these three proteins, uS2-uS5-eS21, on the pre-40S. This will let the readers understand the relative positions and interactions and know how they are loaded as a cluster.

RESPONSE:  The loading of the uS2-uS5-eS21 cluster into the human pre-40S particle is nicely demonstrated in the study by Cheng et al. (NAR 2022, 50: 11924). We think that adding a figure showing the position of these three ribosomal proteins in the 40S subunit, although interesting, is secondary to the focus of the review article and is redundant to the elegant structures presented in the Chang et al. study.

  1. The color labeling in Fig 2 should be consistent in A and C, making it easier for the readers to compare.

(A) N- and C-terminal TYPP 204 domains (highlighted in red and blue, respectively)

(C) Blue: N-terminal TYPP domain, Red: C-terminal TYPP domain

RESPONSE: We thank the reviewer for noting this oversight. The color labeling in Fig. 2A and 2C are consistent, but the legend to Figure 2C was incorrect:  Red is the N-terminal TYPP whereas the blue is the C-terminal TYPP, as shown in Fig. 2A.  The legend to Fig. 2C was corrected (line 233).

  1. The data in Fig 3 and Fig 5 are generated by AlphaFold-Multimer in this manuscript. More than one prediction model should be given from the program. Thus, please indicate the detailed information and factors of the selective structure.

RESPONSE:  The reviewer raises a good point, and we agree that information on the model selection was lacking. Indeed, the review is correct that AlphaFold-Multimer generates more than one predicted model. Each model produces a per-residue estimate of its confidence on a scale from 0 – 100. This confidence measure is called pLDDT (predicted Local Distance Difference Test) and corresponds to the model’s predicted score. For the models presented in Figures 3 and 5, we chose the best confident relaxed structures, with the highest pLDDT. Importantly, the alternative models showed only few decimals’ differences relative to the best model, indicating uniformity in the predicted structures. As suggested, we have now added such clarifications in the revised manuscript (see lanes 264-267 and 454-456).

  1. Fig 5A: The label of the figure should be changed to 180-degree.

RESPONSE: Corrected.  We thank the review for noting this oversight.